

# Parallelized Domain Decomposition for Multi-Dimensional Lagrangian Random Walk, Mass-Transfer Particle Tracking Schemes

Lucas Schauer[1], Michael J. Schmidt[2], Nicholas B. Engdahl[3], Stephen D. Pankavich[1], David A. Benson[4], and Diogo Bolster[5]

[1]Department of Applied Mathematics and Statistics, Colorado School of Mines, Golden, CO, 80401, USA
[2]Center for Computing Research, Sandia National Laboratories, Albuquerque, NM 87185, USA
[3]Department of Civil and Environmental Engineering, Washington State University, Pullman, WA, 99164, USA
[4]Hydrologic Science and Engineering Program, Department of Geology and Geological Engineering, Colorado School of Mines, Golden, CO, 80401, USA
[5]Department of Civil and Environmental Engineering and Earth Sciences, University of Notre Dame, Notre Dame, IN, 46556, USA

**Correspondence:** Lucas Schauer (lschauer@mines.edu)

**Abstract.** Lagrangian particle tracking schemes allow a wide range of flow and transport processes to be simulated accurately, but a major challenge is numerically implementing the inter-particle interactions in an efficient manner. This article develops a multi-dimensional, parallelized domain decomposition (DDC) strategy for mass-transfer particle tracking (MTPT) methods in which particles exchange mass dynamically. We show that this can be efficiently parallelized by employing large numbers of CPU cores to accelerate run times. In order to validate the approach and our theoretical predictions we focus our efforts on a well known benchmark problem with pure diffusion, where analytical solutions in any number of dimensions are well established. In this work, we investigate different procedures for "tiling" the domain in two and three dimensions, (2-$d$ and 3-$d$), as this type of formal DDC construction is currently limited to 1-$d$. An optimal tiling is prescribed based on physical problem parameters and the number of available CPU cores, as each tiling provides distinct results in both accuracy and run time. We further extend the most efficient technique to 3-$d$ for comparison, leading to an analytical discussion of the effect of dimensionality on strategies for



implementing DDC schemes. Increasing computational resources (cores) within the DDC method produces a trade-off between inter-node communication and on-node work. For an optimally subdivided diffusion problem, the 2-$d$ parallelized algorithm achieves nearly perfect linear speedup in comparison with the serial run up to around 2700 cores, reducing a 5-hour simulation to 8 seconds, while the 3-$d$ algorithm maintains appreciable speedup up to 1700 cores.

## 1 Introduction

Numerical models are used to represent physical problems that may be difficult to observe directly (such as groundwater flow), or that may be tedious, expensive or even impossible to currently study via other methods. In the context of groundwater flow, for example, these models allow us to portray transport in heterogeneous media and bio-chemical species interaction, which are imperative to understanding a hydrologic system's development (e.g., (Dentz et al., 2011; Perzan et al., 2021; Steefel et al., 2015; Scheibe et al., 2015; Tompson et al., 1998; Schmidt et al., 2020b; Li et al., 2017; Valocchi et al., 2019)). Since geological problems frequently require attention to many separate, yet simultaneous processes and corresponding physical properties, such as local mean velocity (advection), velocity variability (dispersion), mixing (e.g., dilution), and chemical reaction, we must apply rigorous methods to ensure proper simulation of these processes. Recent studies (e.g., (Benson et al., 2017; Bolster et al., 2016; Sole-Mari et al., 2020)) have compared classical Eulerian (e.g., finite-difference or finite-element) solvers to newer Lagrangian methods and shown the relative advantages of the latter. Therefore, in this manuscript we explore several approaches to parallelize a Lagrangian method that facilitate the simulation of the complex nature of these problems. Given that all of the complex processes noted above must ultimately be incorporated and that this is the first rigorous study of this kind, we focus on well-established and relatively simple benchmark problems with analytical solutions to derive a rigorous approach to this parallelization.

Lagrangian methods for simulating reactive transport continue to evolve, providing both increased accuracy and accelerated efficiency over their Eulerian counterparts by eliminating numerical dispersion (see (Salamon et al., 2006)) and allowing direct simulation of all subgrid processes (Benson et al., 2017; Ding et al., 2017). Simulation of advection and dispersion (without reaction)



in hydrogeological problems began with the Lagrangian random walk particle tracking (RWPT) algorithm that subjects an ensemble of particles to a combination of velocity and diffusion processes (LaBolle et al., 1996; Salamon et al., 2006). Initially, chemical reactions were added in any numerical time step by mapping particle masses to concentrations via averaging over Eulerian volumes,

then applying reaction rate equations, and finally mapping concentrations back to particle masses for RWPT (Tompson and Dougherty, 1988). This method clearly assumes perfect mixing within each Eulerian volume because subgrid mass and concentration perturbations are smoothed (averaged) prior to reaction. The subsequent over-mixing was recognized to induce a scale-dependent apparent reaction rate that depended on the Eulerian discretization (Molz and Widdowson, 1988;

Dentz et al., 2011), thus eliminating some of the primary benefits of the Lagrangian approach. In response, a method that would allow reactions directly between particles was devised and implemented (Benson and Meerschaert, 2008).

Early efforts to directly simulate bimolecular reactions with RWPT algorithms (Benson and Meerschaert, 2008; Paster et al., 2014) were originally founded on a birth-death process that calculated two prob-

abilities: one for particle-particle collocation and a second for reaction and potential transformation or removal given collocation (i.e. particles that do not collocate cannot react, thus preserving incomplete mixing). The next generation of these methods featured a newer particle-number-conserving reaction scheme. This concept, introduced by (Bolster et al., 2016), and later generalized (Benson and Bolster, 2016; Schmidt et al., 2019; Sole-Mari et al., 2019), employs kernel-

weighted transfers for moving mass between particles, where the weights are equivalent to the above-mentioned collision probabilities under certain modeling choices. These algorithms preserve the total particle count, and we refer to them as mass transfer particle tracking (MTPT) schemes. These particle-conserving schemes address low-concentration resolution issues that arise spatially when using particle-killing techniques (Paster et al., 2013; Benson et al., 2017). Furthermore, MTPT

algorithms provide a realistic representation of solute transport with their ability to separate mixing and spreading processes (Benson et al., 2019). Specifically, spreading processes due to small-scale differential advection may be simulated with standard random walk techniques (LaBolle et al., 1996), and true mixing-type diffusive processes may be simulated by mass transfers between particles. MTPT techniques are also ideally suited to and provide increased accuracy for complex



systems with multiple reactions (Sole-Mari et al., 2017; Engdahl et al., 2017; Benson and Bolster,
2016; Schmidt et al., 2020b), but they are computationally expensive because nearby particles must
communicate. This notion of nearness is discussed in detail in Section 3.

    The objective of this study is to develop efficient, multi-dimensional parallelization schemes
for MTPT-based reactive transport schemes. We conduct formal analyses to provide cost bench-
marks and to predict computational speedup for the MTPT algorithm, both of which to date were
only loosely explored in the 1-d case (Engdahl et al., 2019). Herein, we focus on an implementa-
tion that uses a multi-CPU environment that sends information between CPUs via Message Passing
Interface (MPI) directives within FORTRAN code. In particular, we focus on the relative compu-
tational costs of the inter-particle mass transfer versus message passing algorithms, because the
relative costs of either depend upon the manner in which the computational domain is split among
processors. These mass-transfer methods may be directly compared to smoothed-particle hydro-
dynamics (SPH) methods, and are equivalent when a Gaussian kernel is chosen to govern the
mass transfers (Sole-Mari et al., 2019). Specifically, this work shares similarities with previous in-
vestigations of parallelized smoothed particle hydrodynamics (SPH) methods (Crespo et al., 2011;
Gomez-Gesteira et al., 2012; Xia and Liang, 2016; Morvillo et al., 2021), but is novel as it tackles
nuances that arise specifically for MTPT approaches. A substantial difference within this work is that
the kernels are based on the local physics of diffusion, rather than a user-defined function chosen for
attractive numerical qualities like compact support or controllable smoothness. This adherence to
local physics allows for increased modeling fidelity, including the simulation of diffusion across
material discontinuities or between immobile (solid) and mobile (fluid) species (Schmidt et al.,
2020a, 2019). In general, the parallelization of particle methods depends on assigning groups of par-
ticles to different processing units. Multi-dimensional domains present many options on how best
to decompose the entire computational domain in an attempt to efficiently use available computing
resources. Along these lines, we compare two different domain decomposition (DDC) approaches.
In the one-dimensional case (Engdahl et al., 2019), the specified domain is partitioned into smaller
subdomains so that each core is only responsible for updating the particles' information inside of
a fixed region, though information from particles in nearby subdomains must be used. Hence, the
first two-dimensional method we consider is a naive extension from the existing one-dimensional



technique (Engdahl et al., 2019) that decomposes the domain into vertical slices along the $x$-axis of the $xy$-plane. This method is attractive for its computational simplicity but limits speedup for large numbers of processors (see Section 7). Our second method decomposes the domain into a "checkerboard" consisting of subdomains that are as close to squares (or cubes) as is possible given the number of cores available. RWPT simulations without mixing often require virtually no communication across subdomain boundaries because all particles act independently in the model. However, MTPT techniques require constant communication along local subdomain boundaries at each time step, which leads to challenges in how best to accelerate these simulations without compromising the quality of solutions. This novel, multi-dimensional extension of parallelized DDC techniques for the MTPT algorithm will now allow for simulation of realistic, computationally-expensive systems in seconds to minutes rather than hours to days. Further, based on given simulation parameters, we provide formal run time prediction analysis that was only hypothesized in previous work and will allow future users to optimize parallelization prior to executing simulations. This manuscript rigorously explores the benefits of our parallelized DDC method while providing guidelines and cautions for efficient use of the algorithm.

## 2    Model Description

An equation for a chemically-conservative, single component system experiencing local mean velocity and Fickian diffusion-like dispersion is

$$\frac{\partial C}{\partial t} + \nabla \cdot (\boldsymbol{v}C) = \nabla \cdot (\mathbf{D}\nabla C), \qquad \boldsymbol{x} \in \Omega \subseteq \mathbb{R}^d, \quad t > 0, \tag{1}$$

where $C(\boldsymbol{x},t)$ [mol $L^{-d}$] is the concentration of a quantity of interest, $\boldsymbol{v}(\boldsymbol{x},t)$ [$LT^{-1}$] is a velocity field, and $\mathbf{D}(\boldsymbol{v})$ [$L^2T^{-1}$] is a given diffusion tensor. Advection-diffusion equations of this form arise within a variety of applied disciplines relating to fluid dynamics (Bear, 1972; Tennekes and Lumley, 1972; Gelhar et al., 1979; Bear, 1961; Aris, 1956; Taylor, 1953). Depending on the physical application under study, various forms of the diffusion tensor may result. Often, it can be separated into two differing components, with one representing mixing between nearby regions of differing concentrations and the other representing spreading from the underlying flow (Tennekes and Lumley,





1972; Gelhar et al., 1979; Benson et al., 2019). This decomposition provides a general splitting of the tensor into

$$\mathbf{D} = \mathbf{D}_{\mathrm{mix}}(\boldsymbol{v}) + \mathbf{D}_{\mathrm{spread}}(\boldsymbol{v}). \tag{2}$$

Lagrangian numerical methods, such as those developed herein, can then be used to separate the simulation of these processes into mass transfer algorithms that capture the mixing inherent to the system and random walk methods that represent the spreading component (see, e.g., (Ding et al., 2017; Benson et al., 2019)). As our focus here is mainly driven by the novel implementation of diffusive processes in MTPT algorithms, we will for now assume a purely diffusive system so that $\boldsymbol{v}(\boldsymbol{x}) = \mathbf{0}$. This assumption results in an isotropic diffusion tensor that reduces to

$$\mathbf{D} = D\mathbb{I}_d, \tag{3}$$

where $\mathbb{I}_d$ is the $d \times d$ identity matrix. The remaining scalar diffusion coefficient can also be separated into mixing and spreading components, according to

$$D = D_{\mathrm{mix}} + D_{\mathrm{spread}}. \tag{4}$$

Despite the assumption of zero advection, we simulate spreading via random walks as an eventual necessity for moving particles within our DDC scheme. Stationary particles do not provide computational complexity for the mass transfer algorithm as distances between particles remain constant.

## 2.1 Initial Conditions and Analytic Solution

We define a general and well-established benchmark test problem to facilitate the analysis of speedup and computational efficiency. Based on the chosen tiling method, the global domain is subdivided into equisized subdomains, and each core knows its own local, non-overlapping domain limits. The particles are then load balanced between the cores and randomly scattered within the local domain limits. To represent the initial mass distribution, we use a Heaviside function in an $L^d$-sized domain, which assigns all particles with position $x \geq L/2$ with mass $M = 1$ and assigns no mass to particles with position $x < L/2$ (i.e. a heaviside step function initial condition). This initial condition will





allow us to assess the accuracy of simulations as, for an infinite domain (simulated processes occur

away from boundaries for all time), it admits an exact analytical solution

$$C(x,t) = \frac{1}{2}\text{erfc}\left[-(x-x')/4Dt\right], \tag{5}$$

where $x' = L/2$ and $t$ is the elapsed time of the simulation. The existence of an analytical solution is beneficial to our ability to rigorously test our proposed schemes. We compare simulated results to this solution using the root-mean-squared error (RMSE). Note that all dimensioned quantities are

unitless for the analysis we conduct and all references to run times are measured in CPU wall clock time.

### 2.2 Simulation Parameters

Unless otherwise stated, all 2-$d$ simulations will be conducted with the following computational parameters: the $L \times L$ domain is fixed with $L = 1000$; the time step is fixed to $\Delta t = 0.1$; the number

of particles is $N = 10^7$; and the diffusion constant is chosen to be $D = 1$. The total time to be simulated is fixed as $T = 10$, which results in 100 time steps during each simulation. We choose parameters in an attempt to construct problems with similar cost across dimensionality. Hence, all 3-$d$ simulations will be conducted with the same diffusion constant and time step size, but they will be in an $L^3$ domain with $L = 100$ and with a number of particles $N = 5 \times 10^6$ . In general, we will

always use a $\Delta t$ that satisfies the optimality condition

$$\min\{\Delta t\} \geq \frac{\left(\frac{L}{\sqrt[d]{N}}\right)^2}{\frac{1}{\beta}2D}, \tag{6}$$

formulated in (Schmidt et al., 2022), in which $\beta$ is a kernel bandwidth parameter described in Section 3.

### 2.3 Hardware Configuration

The simulations in this paper were performed on a heterogeneous High Performance Computing (HPC) cluster called Mio, which is housed at Colorado School of Mines. Each node in the cluster contains 8-24 cores, with clock speeds ranging from 2.50-GHz to 3.06-GHz. Mio uses a network





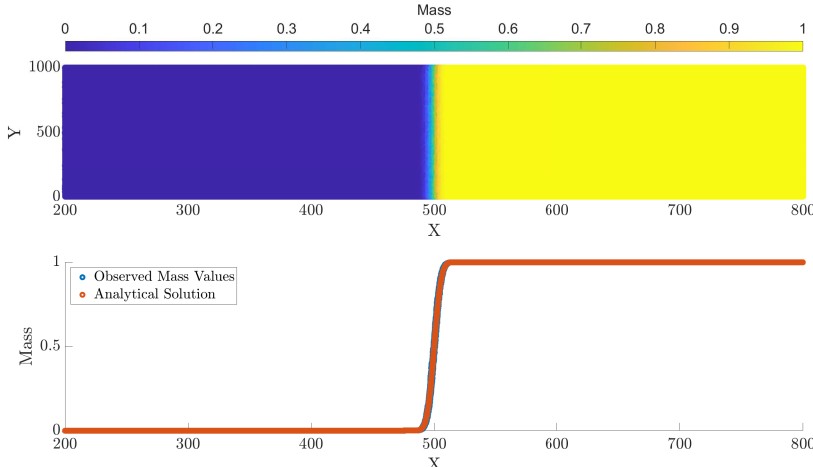

**Figure 1.** The top figure displays the computed particle masses at final simulation time $T = 10$, and the bottom figure provides a computed vs. analytical solution comparison at the corresponding time. The parameters for this run are $N = 10^7$, $\Delta t = 0.1$, and $D = 1$.

of Infiniband switches to prevent performance degradation in multinode simulations. We use the compiler `gfortran` with optimization level 3, and the results we present are averaged over an
ensemble of 5 simulations to reduce noise that is largely attributable to the heterogeneous computing architecture.

## 3   Mass Transfer Particle Tracking Algorithm

The MTPT method simulates diffusion by weighted transfers of mass between nearby particles. These weights are defined by the relative proximity of particles that is determined by construct-
ing and searching a K-D tree (Bentley, 1975). Based on these weights, a sparse transfer matrix is created that governs the mass updates for each particle at a given time step. As previously noted (Schmidt et al., 2018), PT methods allow the dispersive process to be simulated in two distinct ways by allocating a specific proportion to mass transfer and the remaining portion to random walks.



Given the diffusion coefficient $D$, we introduce $\kappa \in [0,1]$ such that

$$D_{\text{RW}} = D_{\text{spread}} = \kappa D \tag{7}$$

and

$$D_{\text{MT}} = D_{\text{mix}} = (1 - \kappa)D. \tag{8}$$

We choose $\kappa = 0.5$ to give equal weight to the mixing and spreading in simulations. Within each time step, the particles first take a random walk in a radial direction, the size of which is based on the value of $D_{RW}$. Thus, we update the particle positions via the first-order expansion

$$X_i(t + \Delta t) = X_i(t) + \xi_i \sqrt{2D_{RW}\Delta t}, \tag{9}$$

where $\xi_i$ $[TL^{-1}]$ is a standard normal Gaussian random variable. We enforce zero-flux boundary conditions, implemented as a perfect elastic collision/reflection when particles random walk outside of the domain. We define a search radius, $\psi$, that is used in the K-D tree algorithm given by

$$\psi = \lambda \sqrt{\frac{1}{\beta} 2D_{MT}\Delta t}, \tag{10}$$

where $\sqrt{\beta^{-1} 2D_{MT}\Delta t}$ is the standard deviation of the mass-transfer kernel, $\Delta t$ is the size of the time step, $D_{MT}$ is the mass-transfer portion of the diffusion coefficient, and $\lambda$ is a user-defined parameter that determines the radius of the search. We choose a commonly-employed value of $\lambda = 6$, as this will capture more than 99.9% of the relevant particle interactions; however, using smaller values of $\lambda$ can marginally decrease run time at the expense of accuracy. Using the neighbor list provided by the K-D tree, a sparse weight matrix is constructed that will transfer mass amongst particles based on their proximity. Since particles are moving via random walks, the neighbor list and corresponding separation distances will change at each time step, requiring a new K-D tree structure and sparse weight matrix within each subdomain. The mass transfer kernel we use is given by

$$K(\boldsymbol{x}_i, \boldsymbol{x}_j) = \frac{1}{\sqrt{(4\pi\beta^{-1}\Delta t)^d \det(\mathbf{D}_{MT})}} \exp\left(-\frac{(\boldsymbol{x}_i - \boldsymbol{x}_j)^T \mathbf{D}_{MT}^{-1}(\boldsymbol{x}_i - \boldsymbol{x}_j)}{4\beta^{-1}\Delta t}\right). \tag{11}$$



Here, $\beta > 0$ is a tuning parameter that encodes the mass transfer kernel bandwidth $h = \sqrt{\frac{1}{\beta} 2 \det(\mathbf{D}_{MT}) \Delta t}$, and we choose $\beta = 1$ hereafter. Recalling $\mathbf{D}_{MT} = D_{MT} \mathbb{I}$ and substituting for the kernel bandwidth $h = \sqrt{2 D_{MT} \Delta t}$, we can simplify the formula in Equation (11) to arrive at

$$K(\boldsymbol{x}_i, \boldsymbol{x}_j) = \frac{1}{(2\pi h^2)^{\frac{d}{2}}} \exp\left( -\frac{\|\boldsymbol{x}_i - \boldsymbol{x}_j\|^2}{2h^2} \right). \tag{12}$$

Next, we denote

$$\mathcal{K}_{ij} = K(\boldsymbol{x}_i, \boldsymbol{x}_j)$$

for each $i, j = 1, ..., N$ and normalize the MT kernel to ensure conservation of mass (Sole-Mari et al., 2019; Herrera et al., 2009; Schmidt et al., 2017). This produces the weight matrix $\mathbf{W}$ with entries

$$W_{ij} = \frac{\mathcal{K}_{ij}}{\frac{1}{2}\left(\sum_{i=1}^{N} \mathcal{K}_{ij} + \sum_{j=1}^{N} \mathcal{K}_{ij}\right)}, \tag{13}$$

that is used in the mass transfer step (15). The algorithm updates particle masses, $M_i(t)$, via the first-order approximation

$$M_i(t + \Delta t) = M_i(t) + \delta_i, \tag{14}$$

where

$$\delta_i = \sum_{j=1}^{N} \beta(M_j(t) - M_i(t)) W_{ij} \tag{15}$$

is the change in mass for a particular particle during a time step. This can also be represented as a matrix-vector formulation by computing

$$\boldsymbol{\delta} = \mathbf{W} M, \tag{16}$$

where $M$ is the vector of particle masses, and then updating the particle masses at the next time step via the vector addition

$$M(t + \Delta t) = M(t) + \boldsymbol{\delta}. \tag{17}$$





225 In practice, imposing the cut-off distance $\psi$ from Equation (10) further implies that $\mathbf{W}$ is sparse and allows us to use a sparse forward matrix solver to efficiently compute the change in mass. Finally, the algorithm can convert masses into concentrations for comparison with the analytic solution using

$$\boldsymbol{C}(t) = \frac{N\boldsymbol{M}(t)}{L^d} \tag{18}$$

in $d$ dimensions with an $L^d$-sized simulation domain.

## 4   Domain Decomposition

With the foundation of the algorithm established, we focus on comparing alternative tiling strategies within the domain decomposition method and their subsequent performance.

### 4.1   Slices Method

The first approach extends the 1-$d$ technique by slicing the 2-$d$ domain along a single dimension, depending on how many cores are available for use. For example, depending on the number of computational cores allocated for a simulation, we define the width of each region as

$$\Delta x = \frac{L}{N_\Omega}, \tag{19}$$

where $N_\Omega$ is the number of subdomains. In addition, we impose the condition that $N_\Omega$ is equal to the number of allocated computational cores. So, the region of responsibility corresponding to the first core will consist of all particles with $x$-values in the range $[x_{\min}, \Delta x)$, and the next core will be responsible for all particles with $x$-values in the interval $[\Delta x, 2\Delta x)$. This pattern continues through the domain with the final core covering the last region of $[(N_\Omega - 1)\Delta x, x_{\max}]$. Each of these slices covers the entirety of the domain in the $y$-direction, so that each core's domain becomes thinner as the number of cores increases. A graphical example of the slices method decomposition is shown in Figure 2(a).



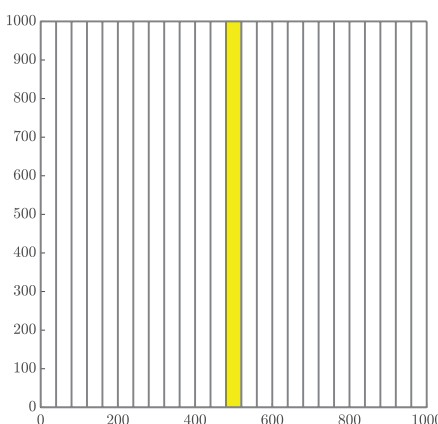

(a) Decomposing the domain with 25 Cores using the Slices Method. Figure 4(a) displays an enlarged slices domain with a description of ghost particle movement, as well.

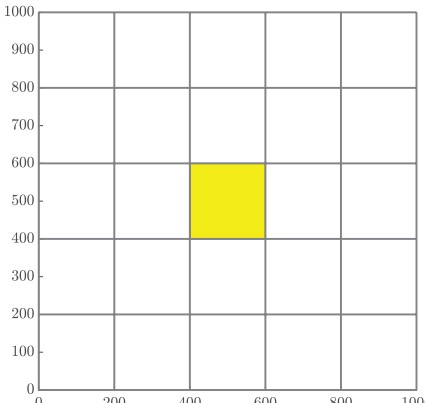

(b) Decomposing the domain with 25 Cores using the Checkerboard Method. Figure 4(b) displays an enlarged checkerboard domain with a description of ghost particle movement, as well.

**Figure 2.** General schematics of (a) slices and (b) checkerboard domain decompositions.





## 4.2 Checkerboard Method

In addition to the slices method, we consider decomposing the domain in a tiled or "checkerboard" manner. Given a $W \times H$ domain (without loss of generality, we assume $W \geq H$), we define $A = W/H$ to be the aspect ratio. Then, choosing $N_\Omega$ subdomains (cores) we determine a pair of integer factors, $f_1, f_2 \in \mathbb{N}$ with $f_1 \leq f_2$, whose ratio most closely resembles that of the full domain, i.e. $f_1 f_2 = N_\Omega$ such that

$$|f_2/f_1 - A| \leq |g_2/g_1 - A| \tag{20}$$

for any other pair $g_1, g_2 \in \mathbb{N}$. Then, we decompose the domain by creating rectangular boxes in the horizontal and vertical directions to most closely resemble squares in 2-$d$ or cubes in 3-$d$. If the full domain is taller than it is wide, then $f_2$ is selected as the number of boxes in the vertical direction. Alternatively, if the domain is wider than it is tall, we choose $f_1$ for the vertical decomposition. If we assume that $W \geq H$ as above, then the grid box dimensions are selected to be

$$\Delta x = \frac{x_{\max} - x_{\min}}{f_2}, \tag{21}$$

and

$$\Delta y = \frac{y_{\max} - y_{\min}}{f_1}. \tag{22}$$

With this, we have defined a grid of subregions that cover the domain, spanning $f_2$ boxes wide and $f_1$ boxes tall to use all of the allocated computational resources. Assuming $N_\Omega$ is not a prime number, this method results in a tiling decomposition as in Figure 2(b). Note that using a prime number of cores reverts the checkerboard method to the slices method.

## 5 Ghost Particles

In MTPT algorithms, nearby particles must interact with each other. Specifically, a particle will exchange mass with all nearby particles within the search radius in Equation (10). Our method of applying domain decomposition results in subdomains that do not share memory with neighboring





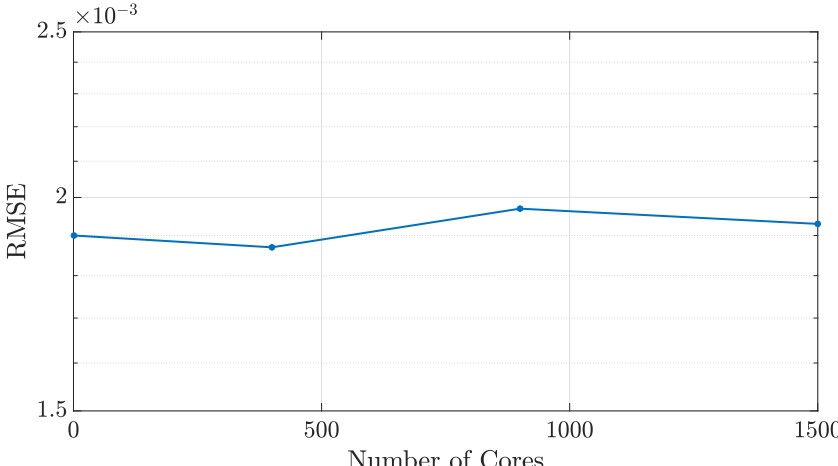

**Figure 3.** The algorithm does not incur noteworthy changes in error, as a function of $N_\Omega$ for the 2-$d$ checkerboard DDC method considered here, nor in any of the simulations that were performed.

regions. If a particle is near a local subdomain boundary, it will require information from particles that are near that same boundary in neighboring subdomains. Thus, each core requires information from particles in a buffer region just outside the core's boundaries, and because of random walks, 270 the particles that lie within this buffer region must be determined at each time step. The size of this buffer zone is defined by the search distance in Eq. (10). The particles inside these buffers are called "ghost" particles and their information is sent to neighboring subdomains' memory using MPI. Because each local subdomain receives all particle masses within a $\psi$-sized surrounding buffer of the boundary at each time step, the method is equivalent to the $N_\Omega = 1$ case after constructing the 275 K-D tree on each subdomain, resulting in indistinguishable nearest-neighbor lists. Although ghost particles contribute to mass transfer computations, the masses of the original particles, to which the ghost particles correspond, are not altered via computations on domains in which they do not reside. Thus, we ensure an accurate, explicit solve for only the particles residing within each local subdomain during each time step.



The process we describe here differs depending on the decomposition method. For example, the slices method gives nearby neighbors only to the left and to the right (Figure 4(a)). On the other hand, the checkerboard method gives nearby neighbors in 8 directions. The communication portion

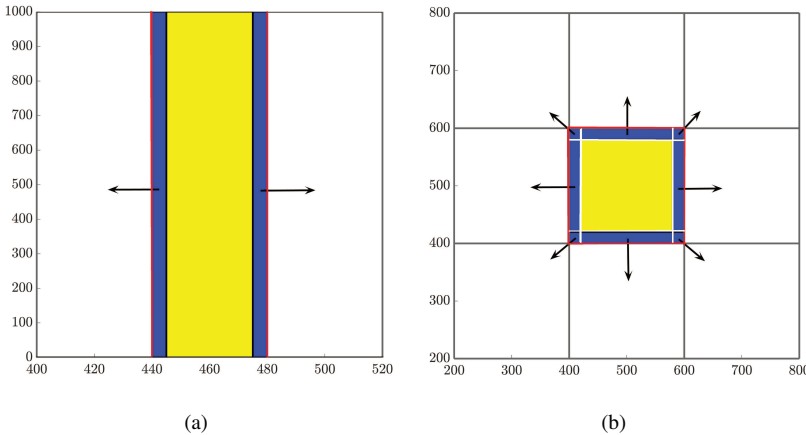

(a)                                    (b)

**Figure 4.** All particles within a buffer width of $\psi$ from the boundary of a subdomain (blue) are sent to the left and to the right for reaction in the slices method (a), whereas they are sent to 8 neighboring regions in the checkerboard method (b). Note that the red lines depict subdomain boundaries, and the black arrows indicate the outward send of ghost particles to neighbors. As well, note that the tails of the black arrows begin within the blue buffer region. Ghost pad size is exaggerated for demonstration.

of the algorithm becomes more complicated as spatial dimensions increase. In 3-$d$, we decompose the domain using a similar method to prescribe a tiling as in 2-$d$, but the extra sends and receives
to nearby cores significantly increase. For example, the 2-$d$ algorithm must pass information to 8 nearby cores, whereas the 3-$d$ algorithm must pass information to 26 nearby cores—8 neighboring cores on the current plane and 9 neighboring cores on the planes above and below.



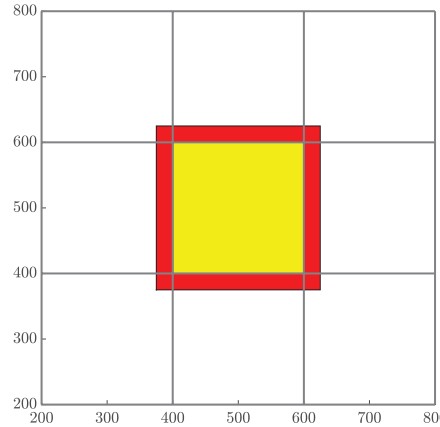

**Figure 5.** The red band represents all particles that will be received by the local subdomain (yellow) from neighboring regions for mass transfer. The number $N_S$ quantifies the number of particles that are involved in the MT step of the algorithm, which is the combination of all particles whose positions are in either the red or yellow region.

# 6 Cost Analysis

## 6.1 Mass-transfer Cost

In this section, we characterize and predict the amount of work being performed within each of portion of the algorithm. The general discussion of work and cost here refer to the run times required within distinct steps of the algorithm. We profile the code that implements the MTPT algorithm using the built-in, Unix-based profiling tool `gprof` (Graham et al., 2004) that returns core-averaged run times for all parent and child routines. The two main steps upon which we focus are the communica-

tion step and the Mass Transfer (MT) step. For each subdomain, the communication step determines which particles need to be sent (and where they should be sent) and then broadcasts them to their correct nearby neighbors. The MT step carries out the interaction process described in Section 3 using all of the particles in a subdomain and the associated ghost particles, the latter of which are





not updated within this process. As these two processes are the most expensive components of the
algorithm, they will allow us to project work expectations onto problems with different dimensions
and parameters.

We begin with an analysis of the MT work. First, in the interest of tractability, we will consider
only regular domains, namely a square domain with sides of length $L$ so that $\Omega_x = \Omega_y = L$ in 2-$d$
and a cubic domain with $\Omega_x = \Omega_y = \Omega_z = L$ in 3-$d$. Hence, the area and volume of these domains
are $A = L^2$ and $V = L^3$, respectively. Also, we define the total number of utilized processors to be
$P$ and take $P = N_\Omega$ so that each subdomain is represented by a single processor. Assuming that $P$
is a perfect $d^{th}$ power and the domain size has the form $L^d$ for dimension $d$, this implies that there
are $P^{1/d}$ subdivisions (or "tiles" from earlier) in each dimension. Further, we define the density
of particles to be $\rho_d = N/L^d$ in $d$-dimensions where $N$ is the total number of particles. Finally,
recall that the pad distance, which defines the length used to determine ghost particles, is defined by
$\psi = 6\sqrt{2D_{MT}\Delta t}$. With this, we let $N_S$ represent the number of particles that will be involved in
the mass transfer process on each core, and this can be expressed as

$$N_S = \rho_d \left( \frac{L}{P^{1/d}} + 2\psi \right)^d = N \left( \frac{1}{P^{1/d}} + \frac{2\psi}{L} \right)^d, \quad d = 1, 2, 3, \tag{23}$$

which is an approximation of the number of particles in an augmented area or volume of each
local subdomain, accounting for the particles sent by other cores. Figure 5 illustrates $N_S$ as the
number of particles inside the union of the yellow region (the local subdomain's particles) and the
red region (particles sent from other cores). As previously mentioned, we construct and search a K-D
tree to find a particles' nearby neighbors. Constructing the tree structure within each subdomain is
computationally inexpensive, and searching the tree is significantly faster than a dense subtraction
to find a particle's nearby neighbors: for $N$ particles in memory, the dense subtraction is $\mathcal{O}(N)$
expensive for a single particle, while the K-D tree search is only $\mathcal{O}(\log_{10}(N))$. Based on the results
from gprof, searching the K-D tree is consistently the most dominant cost in the MT routine. As
a result, the time spent in the mass transfer routine will be roughly proportional to the speed of
searching the K-D tree. This approximation results in the MT costs scaling according to (Kennel,
325  2004):

$$T_S = \alpha_d N_S \log(N_S), \tag{24}$$



where $\alpha_d$ is a scaling coefficient reflecting the relative average speed of the calculations per particle for $d = 2, 3$. Note that $\alpha_3 > \alpha_2$, as dimension directly impacts the cost of the K-D tree construction. We are able to corroborate this scaling for both 2-$d$ and 3-$d$ problems by curve fitting to compare

$N_S \log(N_S)$ for each method of DDC to the amount of time spent in the MT subroutine. In particular, we analyzed the empirical run time for the K-D tree construction and search in an ensemble of 2-$d$ simulations with the theoretical cost given by (24). Figures 6(a) and 6(b) display the run times plotted against our predictive curve for the MT portion of the algorithm, exhibiting a coefficient of determination ($r^2$) close to 1.

Note that changing the total number of particles within a simulation should not change the scaling relationship as Equation (24) only depends on the value of $N_S$. This relationship implies that a simulation with a greater number of particles and using a greater number of processors can achieve the same value of $N_S$ as a simulation with fewer total particles and processors. When the values of $N_S$ coincide across these combinations of particles and processors, we expect the time

spent in the MT subroutine to be the same (Fig. 7). We see that our predictions for the MT subroutine, based on proportionality to the K-D tree search, provide a reliable run time estimate in both the 2-$d$ and 3-$d$ cases. We also observe an overlay in the curves as $N$ increases, which directly increases the amount of work for the MT portion of the algorithm. For instance, if we consider a range of particle numbers in both dimensions (Fig. 7), we see the respective curves exhibit similar run time

behavior as $N_S$ decreases. The plots of MT run time display an approximately linear decrease as $N_S$ decreases, which would seem to indicate continued performance gains with the addition of more cores. However, one must remember that adding cores is an action of diminishing returns because the local core areas or volumes tend to zero as more are added, and $N_S$ tends to a constant given by the size of the surrounding ghost particle area (see Fig. 5). For example, Figure 6(a) shows

that MT run time only decreases by around half of a second from adding nearly 1500 processors. Predictions concerning this tradeoff are made in Section 7. In particular, there appears to be a similar end behavior in both 2-$d$ and 3-$d$ as the number of cores is increased (so that $N_S$ is decreased), which can be attributed mostly to the asymptotic nature of $N_S$ shown in Figure 8.



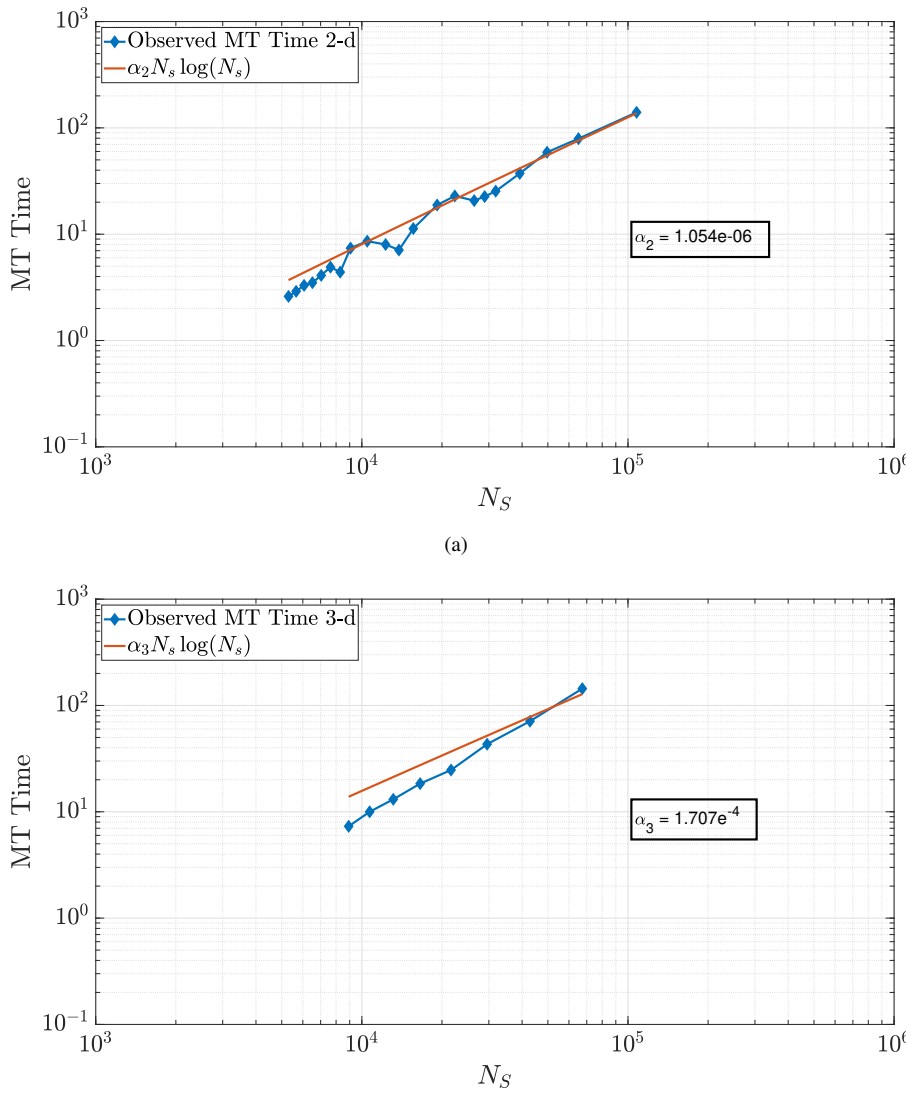

**Figure 6.** Plots of runtime in the MT portion of benchmark runs. Note the similar behavior in both 2-$d$ (a) and 3-$d$ (b) for predicting MT subroutine run time, based on our theoretical run time scaling in Equation (24). Using this prediction function achieves values of $r^2 = 0.9780$ in 2-$d$ and $r^2 = 0.9491$ in 3-$d$. Axis bounds are chosen for ease of comparison to results in Figures 7 and 9.

 

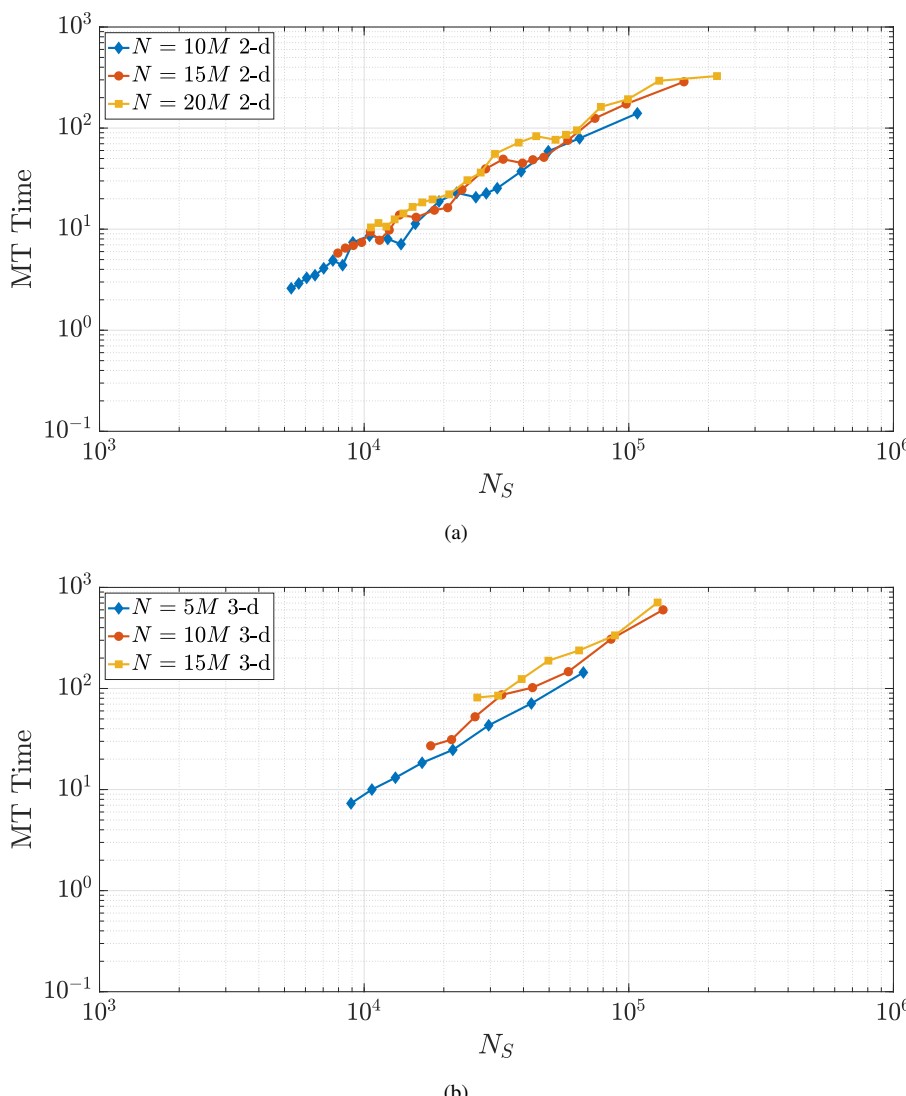

(a)

(b)

**Figure 7.** Varying the total particle number $N$ directly influences the value of $N_S$, as the right-most side of Equation (23) reflects. Simulations across these different values of $N$ in (a) 2-$d$ and (b) 3-$d$ exhibit common behavior with respect to MT run time as $N_S$ decreases. Axis limits are chosen for comparison with Figure 9.



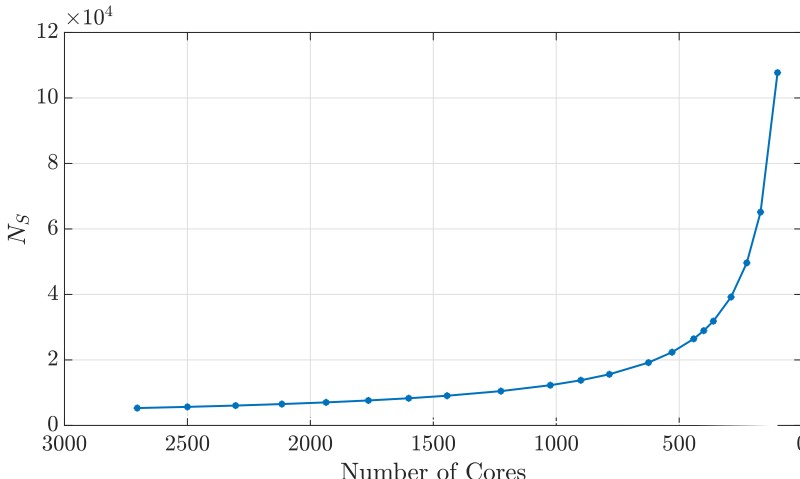

**Figure 8.** As cores increase, $N_S$ decreases to a constant, resulting in a constant amount of work in the MT subroutine.

## 6.2 Ghost Particle Communication Cost Analysis

Next, we explore the time required to send and receive ghost particle information between subdomains within the MPI-based communication subroutine. This time includes three processes on each core: evaluating logical statements to determine which particles to send to each neighboring core, sending the particles to the neighboring cores, and receiving particles from all neighboring cores. Here, we encounter the issue of load balancing, namely the process of distributing traffic so that cores with less work to do will not need to wait on those cores with more work. Hence, we only need to focus our projections on the cores that will perform the greatest amount of work. These cores (in both dimensions) are the "interior" subdomains, or the subdomains with neighbors on all sides. In 2-$d$ these subdomains will receive particles from 8 neighboring domains, with 4 neighbors sharing edges and 4 on adjacent corners. In 3-$d$, particles are shared among 26 neighboring subdomains. Similar to the MT analysis, we observe that both the 2-$d$ and 3-$d$ data in Figure 9 exhibit similar curves across varying particle numbers, respectively, as the number of processors becomes large (i.e., as $N_S$ becomes small). This eventual constant cost is to be expected in view of the asymp-





totic behavior of $N_S$ as $P$ grows large. Note also that the 3-$d$ MPI simulation times are consistently around 5 to 10 times greater than 2-$d$ because of the increased number of neighboring cores involved
in the ghost particle information transfer.

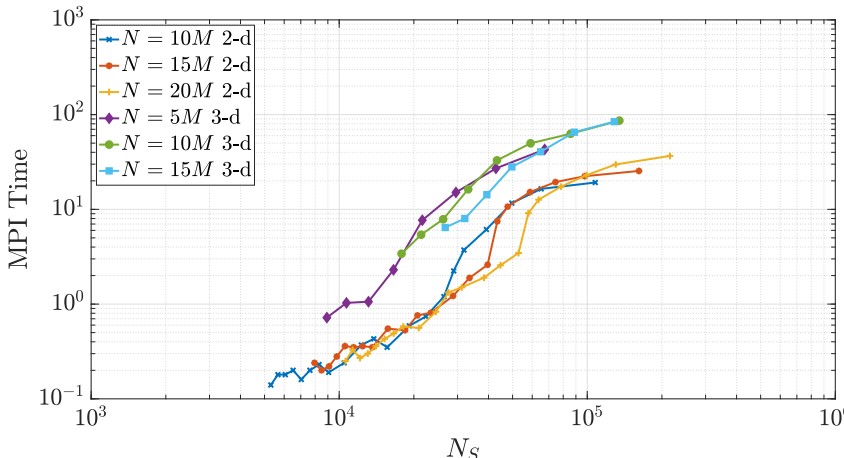

**Figure 9.** As $N_S$ decreases, we observe similar trends in the MPI subroutine run time in both 2-$d$ and 3-$d$, respectively. The 3-$d$ MPI times are, in general, 5 to 10 times slower than similar runs in 2-$d$.

## 7 Speedup Results

In this section, we discuss the advantages and limitations of each method by evaluating the manner in which the decomposition strategies accelerate run times. We employ the quantity "speedup" in order to compare the results of our numerical experiments. The speedup of a parallelized process is
commonly formulated as

$$S_P = \frac{T_1}{T_P}, \tag{25}$$

where $T_P$ is the run time using $P$ cores and $T_1$ is the serial run time. We also use the notion of efficiency that relates the speedup to the number of processors, and is typically formulated as

$$E_P = \frac{S_P}{P}. \tag{26}$$





If the parallelization is perfectly efficient, then $P$ cores will yield a $P$ times speedup from the serial run, producing a value of $E_P = 1$. Hence, we compare speedup performance to establish a method
that best suits multi-dimensional simulations.

We may also construct a theoretical prediction of the expected speedup due to the runtime analysis of the preceeding section. First, assume that the subdomains are ideally configured as squares in 2-$d$ or cubes on 3-$d$. In this case, the MT runtimes always exceed the MPI times. For smaller values of $N_S$, the MT runtimes are approximately 10 to 100 times larger than those of the MPI step.
Furthermore, the larger MT times are approximately linear with $N_S$ over a large range, regardless of total particle numbers and dimension. Therefore, we may assume that the runtimes are approximately linear with $N_S$ and compare runtimes for different values of $P$. Specifically, for a single processor, all of the particles contribute to the MT runtime, so the speedup can be calculated using equation (23) in the denominator:

$$S_P = \frac{N}{N_S} = \frac{N}{N\left(\frac{1}{P^{1/d}} + \frac{2\psi}{L}\right)^d} = \frac{1}{\left(\frac{1}{P^{1/d}} + \frac{2\psi}{L}\right)^d}. \tag{27}$$

Now, letting $\mathcal{E} \in (0, 1)$ represent a desired efficiency threshold, we can identify the maximum number of processors that will deliver an efficiency of $\mathcal{E}$ based on the size $L$ of the domain, the physics of the problem, and the optimal timestep $\Delta t$ that defines the size of the ghost region (given in terms of the pad distance $\psi$). In particular, using the above efficiency formula, we want

$$\mathcal{E} \le E_P = \frac{S_P}{P} = \frac{1}{P\left(\frac{1}{P^{1/d}} + \frac{2\psi}{L}\right)^d}. \tag{28}$$

A rearrangement then gives the inequality

$$P \le \frac{1}{\mathcal{E}}\left(\frac{(1 - \mathcal{E}^{1/d})L}{2\psi}\right)^d, \tag{29}$$

which provides an upper bound on the suggested number of processors to use once $\psi$ is fixed and a desired minimum efficiency is chosen.
This gives the user a couple of options before running a simulation. The first option is to choose a desired minimum efficiency and obtain a suggested number of processors to use based on the inequality in (29). This option is ideal for users who request or pay for the allocation of computational





resources and must know the quantity of resources to employ in the simulation. The second option is to choose a value for the number of processors and apply the inequality (28) to obtain an estimated efficiency level for that number of processors. This second case may correspond to users who have free or unrestricted access to large amounts of computing resources and may be less concerned about loss of efficiency.

Using Equation (27), we can predict speedup performance for any simulation once the parameters are chosen. The speedup prediction inequalities from above depend only on the domain size and the search distance $\psi$. From these inequalities, the effects of dimensionality while implementing DDC can be conceptualized in two ways. First, if the hypervolume is held constant as dimension changes, particle density also remains constant, which should generally not induce memory issues moving to higher dimensions. This requires choosing a desired hypervolume $\mathcal{V}$ and then determining a length scale along a single dimension with $L = \mathcal{V}^{1/d}$. Figure 10(a) displays speedup predictions for 1-$d$, 2-$d$, and 3-$d$ simulations in domains with equal hypervolumes and fixed $D = 1$ and $\Delta t = 0.1$. Keeping hypervolume constant shows the cost of complexity with increasing dimensions, which reduces efficiency at larger amounts of cores. Conversely, a physical problem may have fixed size on the order of $L^3$, and a user may wish to perform upscaled simulations in 1-$d$ and 2-$d$ before running full 3-$d$ simulations. Figure 10(b) shows the opposite effect: for a fixed length scale $L$, the lower-dimension simulations suffer degraded efficiency for lower number of cores.

The 2-$d$ and 3-$d$ benchmark simulations used in previous sections allow us to calculate both the empirical (observed) and theoretical speedups, and the overlays in Figure 11 and Figure 12 show reasonably accurate predictions over a large range of core numbers. The observed run times were averaged over an ensemble of 5 simulations in order to decrease noise. If the checkerboard method is used to decompose the domain, significantly more cores can be used before the inequality (29) is violated for a chosen efficiency. In particular, if we choose a sequence of perfect square core numbers for the a $1000 \times 1000$ domain, nearly linear speed up is observed for over $1000$ cores, and a maximum of $1906$ times speedup at $2700$ cores, the largest number of CPU cores to which we had access. For reference, the $1906$ times speedup performs a 5-hour serial run in 8 seconds, representing around $0.04\%$ of the original computational time.

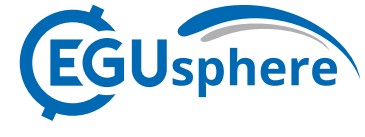

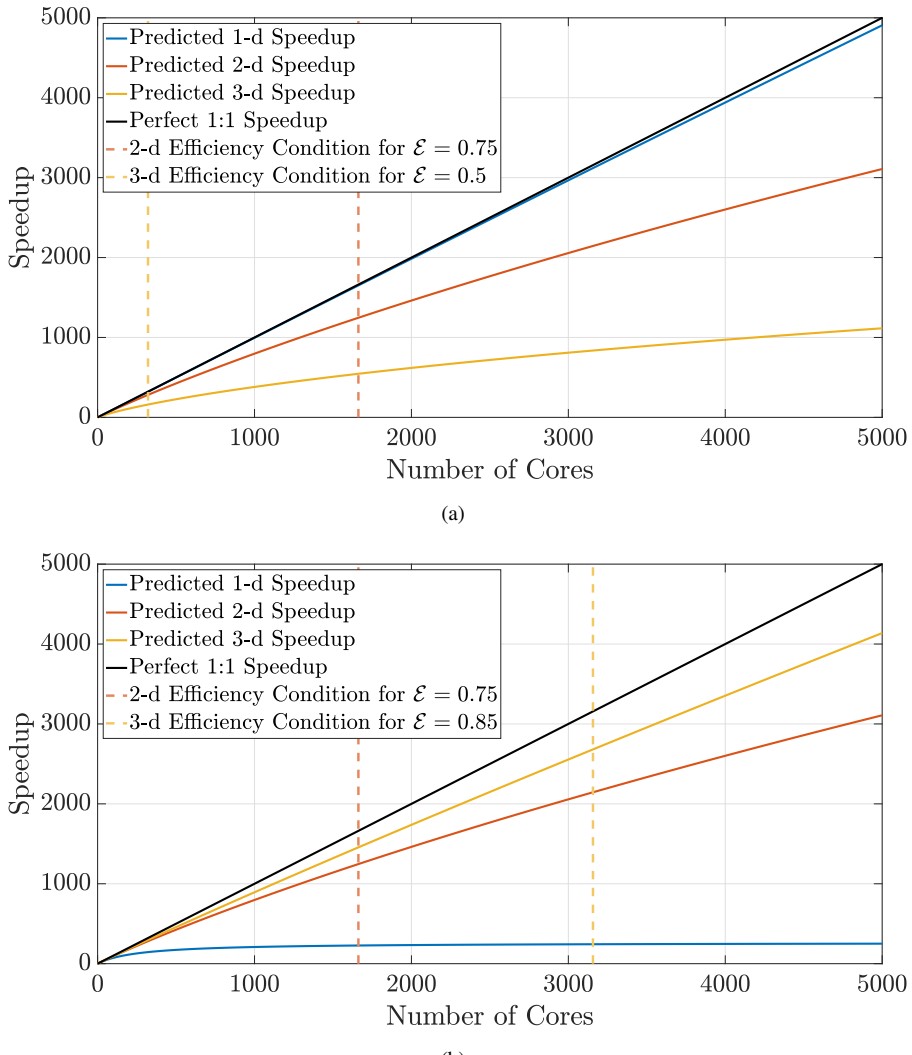

**Figure 10.** The prediction curves give the user a concrete guideline to determine how many cores to allocate for a simulation before performance degrades. The curves in (a) are generated for the defined search distance $\psi$ in a domain with a constant respective hypervolume $\mathcal{V}$, which implies that each dimension's length scale is $L = \mathcal{V}^{1/d}$. The curves in (b) are generated for the same search distance $\psi$ but with $L^d$-sized domains for fixed length $L$.





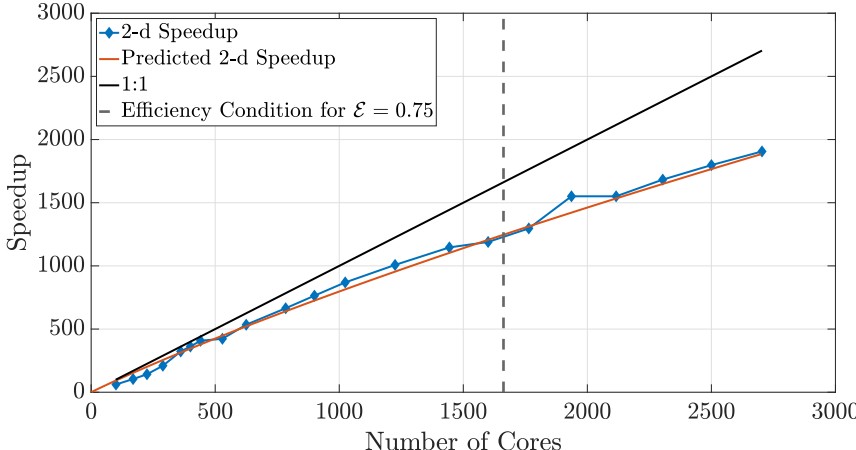

**Figure 11.** Observed (diamonds) and theoretical speedup for 2-$d$ simulations. Each chosen number of processors is a perfect square so that the checkerboard method gives square subdomains. With the chosen parameters $L = 1000, D = 1, \Delta t = 0.1$ and a desired efficiency of 0.75, the upper bound given by the inequality (29) is not violated for the checkerboard method until around 1700 cores.

Finally, we briefly consider the slices method, as it has drastic limitations in 2-$d$ and 3-$d$. Increasing the number of cores used in a simulation while $\psi$ remains fixed causes the ghost regions (as pictured in Figure 4(a)) to comprise a larger ratio of each local subdomain's area. Indeed, if each subdomain sends the majority of its particles, we begin to observe decreased benefits of the parallelization. An inspection of Figure 4(a) suggests that the slices method in 2-$d$ will scale approximately like a 1-$d$ system, because the expression for $N_S$ (Equation (23)) is proportional to the 1-$d$ expression. Indeed, the slices method speedup is reasonably well predicted by the theoretical model for a 1-$d$ model (Figure 13). Furthermore, because the slices method only decomposes the domain along a single dimension, it violates the condition given in (29) at lesser numbers of cores than for the checkerboard method. In fact, using too many cores with the slices method can cease necessary communication altogether once a single buffer becomes larger than the subdomain width. For the given parameter values, this phenomenon occurs at 500 cores with the slices method, so we do not include simulations beyond that number of cores. The speedup for the slices method up to

435

440





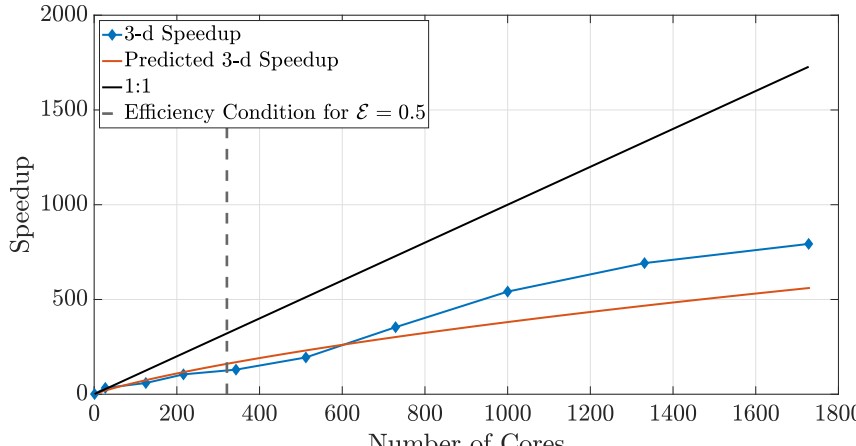

**Figure 12.** Observed (diamonds) and theoretical speedup for 3-$d$ simulations. Each chosen number of processors is a perfect cube so that the checkerboard method gives cubic subdomains. With the chosen parameters $L = 100, D = 1, \Delta t = 0.1$ and a desired efficiency of 0.5, the upper bound given by the inequality (29) is not violated for the checkerboard method until around 320 cores.

500 cores is shown in Figure 13. Although the algorithm is accurate up to 500 cores, we see that
performance deteriorates quite rapidly after around 100 cores, which motivated the investigation of
the checkerboard decomposition.

## 7.1 Non-Square Tilings and Checkerboard Cautions

Given some fixed number of processors (hence subdomains), it is clear that using a subdomain tiling
that is as close as possible to a perfect square (or cube) maximizes efficiency. This occurs when the
factors for subdivisions in each dimension are chosen to most closely resemble the aspect ratio of
the entire domain (shown in 2-$d$ in (20)). Square or cubic subdomains are the most efficient shape
to use, and result in improved speedup that extends to larger numbers of cores. The converse of this
principle means that a poor choice of cores (say, a prime number) will force a poor tiling, and so
certain choices for increased core numbers can significantly degrade efficiency. Figure 14 depicts
results in 2-$d$ for core numbers of $P = 698$ (with nearest integer factors of 2 and 349) and $P = 1322$



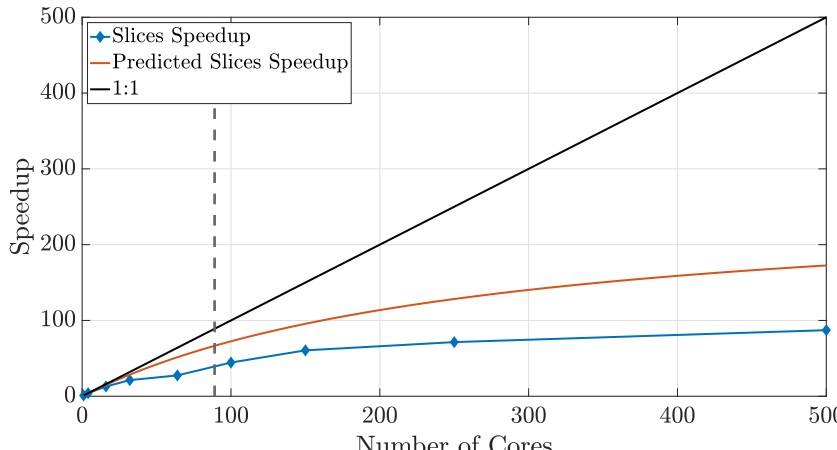

**Figure 13.** Speedup for the slices method plateaus quickly, as the ghost regions increase in proportion to the local subdomain's area.

(with nearest integer factors of 2 and 661) along with well-chosen numbers of cores, namely the perfect squares $P = 400$ and $P = 1600$. It is clear from the speedup plot that simulations with poorly chosen numbers of cores do not yield efficient runs relative to other choices that are much closer to the ideal linear speedup. In particular, we note that the speedup in the case of nearly prime numbers of cores is much closer to the anticipated 1-$d$ speedup. This occurs due to the subdomain aspect ratio being heavily skewed and therefore better resembling a 1-$d$ subdomain rather than a regular (i.e., square) 2-$d$ region.

## 7.2 Non-Serial Speedup Reference Point

We can loosely describe the standard definition of speedup as the quantitative advantage a simulation performed with $P$ cores displays over a simulation running with just a single core. However, a serial run does not require particles to be sent to neighboring regions. Hence, a simulation on a single core does not even enter the MPI subroutine necessary for sending ghost particles, which represents a significant cost. This is not problematic, but it does cast some doubt as to whether the



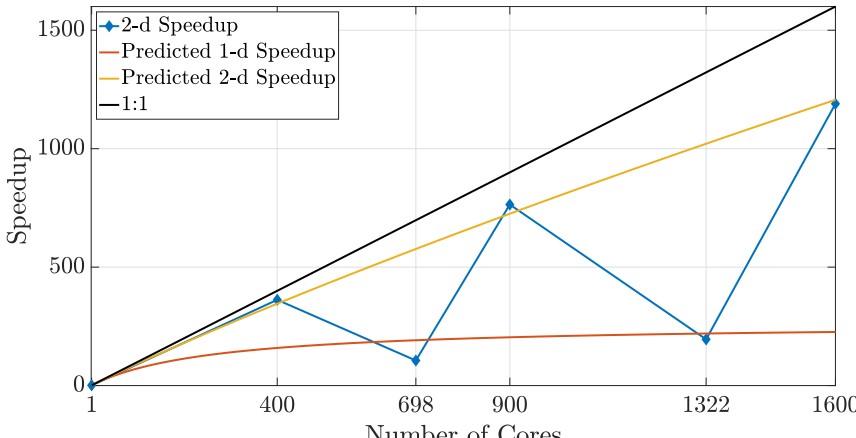

**Figure 14.** Poorly-chosen core numbers may result in severely non-square tilings that can degrade speedup performance, despite employing more computational resources.

470  single core serial case is a reasonable baseline reference for multi-core simulations. For instance, we can compare our speedup results to the 100-core simulation as a reference to observe the reduced computational time incurred by adding cores to an already-parallelized simulation. This provides a different vantage point to measure how well the DDC algorithm performs and can certainly be useful in cases of significantly large particle numbers where a simulation cannot be conducted on less than

475  100 cores due to memory constraints. The plot depicting standard speedup for the checkerboard tiling, which compares all run times to the serial run, is shown in Figure 11. Alternatively, a speedup plot that compares all simulation times to their respective 100-core run times is given in Figure 15. More specifically, this figure displays the speedup ratio given by

$$S_{P_{100}} = \frac{T_{100}}{T_P}, \tag{30}$$

480  where $T_P$ is the run time on $P$ cores and $T_{100}$ is the run time on 100 cores. For example, with 2500 cores, perfect speedup would be 25 times faster than the 100 core run.

The performance in Figure 15 displays above-perfect efficiency for up to 2700 cores, which shows that memory-restricted simulations using very large particle numbers (i.e., the 15M and 20M





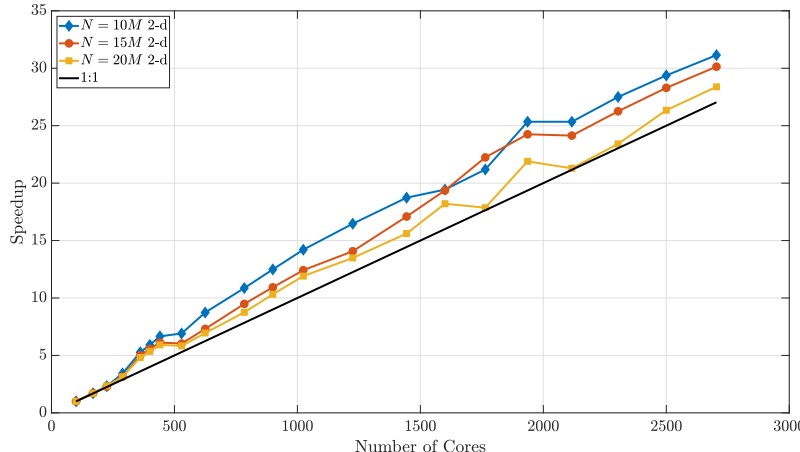

**Figure 15.** A speedup reference point of $T_{100}$ results in super-linear speedup across multiple particle numbers.

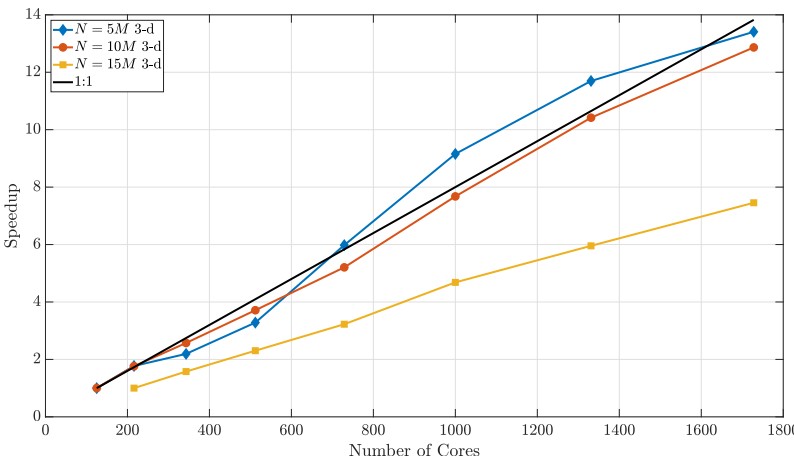

**Figure 16.** Speedup reference points of $T_{125}$ (and $T_{216}$ for the 15M run) result in super-linear speedup for only the $N = 5M$ case, further exemplifying the disparity between 2-d and 3-d.



particle data) can be effectively parallelized to much greater numbers of cores. However, Figure 16
further shows the effect of dimensionality on this comparison, as only those simulations with smaller
particle numbers in 3-$d$ achieve above-perfect efficiency.

## 8   Conclusions and Final Remarks

The checkerboard decomposition for the parallelized DDC algorithm provides significant speedup
to Lagrangian MTPT simulations in multiple dimensions. For a range of simulations, we find that
the mass transfer step is the dominant cost in terms of run time. The approximate linearity of run
time with $N_S$ (defined as total number of native particles and external ghost particles on a single
core/subdomain for mass transfer) allows us to calculate a theoretical speedup that matches empirical
results from well-designed DDC domains. The theoretical predictions also allow one to choose an
efficiency and calculate the optimal number of processors to use, based on the physics of the problem
(specifically, in the context of the benchmarks, we explore domain size $L$, diffusion coefficient $D$,
and time step $\Delta t$). As noted in Section 7, these predictions provide users with a range of needs and
resources necessary forecasting ability that is required before running a large-scale HPC simulation.

Given that we assume a purely-diffusive, non-reactive system in this paper, a necessary extension of this will be an investigation of the performance of these DDC techniques upon adding
advection, reactions, or both to the system. The benchmarks we establish in this work provide accurate expectations for properly load-balanced problems as we begin to add these complexities into
the simulations. In particular, in future efforts we expect to handle variable velocity fields with an
adaptable, moving DDC strategy similar to the technique that we implement in this manuscript. This
initial study is an essential prerequisite before other such advances can even begin. Further, using
local averaging and interpolation of the corresponding velocity-dependent dispersion tensors, we
retain accurate representation of small-scale spreading and mixing despite the subdivided simulation domain. We expect that adequately load balanced, advective-reactive simulations in the future
will exhibit as good as or even better scaling than we have observed in these current benchmarks.
Moving particles via advection is a naturally parallel computation and will only incur minor computational cost as particles move across subdomain boundaries. Moreover, since the general form of our





algorithm simulates complex chemical reactions on particles after mass transfer takes place for all species, we change the most computationally expensive process (the reactions) into a naturally parallel process. Simulating these extra physical phenomena strictly increases computational complexity, but it will be in addition to the computations we carry out here. Given that we are building on these

computations, we fully intend to preserve the predictive ability in more complex simulations moving forward. Finally, another natural extension could address various computational questions by exploring how parallelized MTPT techniques might be employed using shared-memory parallelism, such as OpenMP, CUDA, or architecture-portable parallel programming models like Kokkos, RAJA, or YAKL (Edwards et al., 2014; Trott et al., 2022; Beckingsale et al., 2019; raj, 2022; Norman, 2022).

As we have noted, sending and receiving particles during each time step is a large cost in these simulations, second only to the creation and search of K-D trees and the forward matrix multiplication for mass transfer. Thus, if we could implement a similar DDC technique without physically transmitting ghost particle information between cores and their memory locations, would we expect to see improved speedup for much larger thread counts? A comparison of simulations on a CPU shared

memory system to those on a GPU configuration would represent a natural next step to address this question. In this case, we predict that the GPU would also yield impressive speedup, but it is unclear as to which system would provide lesser overall run times given the significant differences in computational and memory architectures between GPU and CPU systems.

In summary, the checkerboard method in 2-$d$ (and 3-$d$) not only allows simulations to be con-
ducted using large numbers of cores before violating the maximum recommended processor condition given in (29), but also boasts impressive efficiency scaling at a large number of cores. Under the guidelines we prescribe, this method achieves almost perfect linear speedup for more than 1000 processors and maintains significant speedup benefits up to nearly 3000 processors. Our work also showcases how domain decomposition and parallelization can relieve memory-constrained simula-

tions. For example, some of the simulations that we conduct with large numbers of particles cannot be performed with fewer cores due to insufficient memory on each core. However, with a carefully chosen DDC strategy, we can perform simulations with particle numbers that are orders of magnitude greater than can be accomplished in serial, thereby improving resolution and providing higher-fidelity results.



*Code availability.* FORTRAN/MPI codes for generating all results in this paper are held in the public repository `doi:https://doi.org/10.5281/zenodo.6975290`.

*Author contributions.* LS is responsible for writing the original draft and creating the software. LS, SP, DBen, and DBol are responsible for conceptualization. LS, NE, and MS are responsible for methodology. DBen and SP are responsible for formal analysis. All authors are responsible for reviewing and editing the manuscript.

*Competing interests.* The authors declare that they have no conflict of interest.

*Acknowledgements.* The authors and this work were supported by the US Army Research Office under Contract/Grant number W911NF-18-1-0338 and by the National Science Foundation under awards EAR-2049687, EAR-2049688, DMS-1911145, and DMS-2107938. Sandia National Laboratories is a multi-mission laboratory managed and operated by the National Technology and Engineering Solutions of Sandia, L.L.C., a wholly
owned subsidiary of Honeywell International, Inc., for the DOE's National Nuclear Security Administration under contract DE-NA0003525. This paper describes objective technical results and analysis. Any subjective views or opinions that might be expressed in the paper do not necessarily represent the views of the U.S. Department of Energy or the United States Government.



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
