# Peer review of "Parallelized Domain Decomposition for Multi-Dimensional Lagrangian Random Walk, Mass-Transfer Particle Tracking Schemes"

_EGUsphere, 2022_

## Author Response (AR1)

**Reviewer Response**

**1. Reviewer Comment:**

This paper highlights a multi-dimensional, parallel domain decomposition method for mass-transfer particle tracking methods. The authors focus on demonstrating the parallel scalability of a two- and three-dimensional "checkerboard" partitioning, and highlight a theoretical analysis and prediction of scalability at high core counts as their main novel contribution. A grand claim of reducing a "5-hour simulation to 8 seconds" is made to highlight the near-linear scalability of the chosen method.

The paper is well written and gives a detailed and concise overview of the problem definition and theoretical context in which the problem is set. A clear description of the particular class of particle tracking algorithm is provided and appropriate comparisons to other related fields are made. The performance analysis and theoretical prediction of scalability, however, neglects certain established principles (for example weak and strong scaling) and attempts to re-derive some well-known performance analysis steps; for example using an arbitrary 100-core baseline to normalise super-linear speed-ups, instead of normalising relative speed-up over memory sockets or nodes.

Unfortunately, however, the lack of any performance quantification of the baseline result, in particular with respect to shared-memory scaling, casts doubt over the achievements presented. Achieving near-perfect scaling is, after all, much easier with an unoptimised baseline. While the authors are clearly aware of the redundant ghost particle communication within a shared memory space, no attempt is made to quantify single-node/single-socket performance. In particular, despite several of the dominant algorithm components being likely to contain memory-intensive operations (sparse matrix solves, KD-tree lookups, potentially redundant communication buffer copies), no attempt is made to measure or quantify memory-bandwidth utilisation, which could potentially explain some of the observed scaling behaviour (eg. the drop-off in Fig. 9 possibly?). To compound this further, several of the performance graphs are plotting wall time against number of particles per partition, rather than time vs number of core/sockets/nodes, which makes reasoning about scaling behaviour counter-intuitive; and the description of the benchmarking hardware is very loose.

Nevertheless, the authors clearly demonstrate that their implementation of the dominant KD-tree lookups scales as N log(N), and that at high core counts their predicted parallel efficiency can be achieved. The theoretical derivation and comparison of parallel efficiency and scaling behaviour and subsequent mapping to achieved results adds value to the paper.

Overall, I find the topic of the paper and its novel contribution, as stated in the introduction, to be relevant for GMD. The paper has strong potential, but requires significant revisions before I can recommend acceptance, in particular due to the quality of the performance analysis. While I appreciate that fitting shared-memory parallelism is beyond the scope of the benchmarking code, I believe the paper would be significantly improved if the authors would establish single-node performance more rigorously, especially with regards to memory bandwidth utilisation, before focusing on scaling behaviour across multiple sockets/nodes and modelling of algorithmic scaling behaviour.

[1] https://blogs.fau.de/hager/archives/5260

*RESPONSE: We greatly appreciate the reviewer's detailed response and comments. The comments bring insightful ideas for improving the clarity of the paper and provide necessary scrutiny of the scientific computing jargon that we use. Namely, we have incorrectly used the term "processor" throughout the manuscript, which has both confused a knowledgeable reader and undermined the results we present. We should (and will) use the term "core," as we are breaking up workload amongst the physical cores that exist on a single node/processor. This was a mistake in terminology on our part, and we hope that this clears up many of the doubts presented by the reviewer. We ensure that the Message Passing Interface (MPI) routines that we use are, in fact, employing individual cores and not processors. Thus, our serial baseline is, indeed, an ensemble-averaged run on one core, which we believe does not undermine our speedup results. Below, all reviewer comments pertaining to specific figure numbers in the old manuscript have been updated to correspond to the figure they were referencing in the new manuscript.*

**2. Detailed list of comments:**

- Overly strong statements, like "reducing a 5-hour simulation to 8 seconds" (pg 2, l. 17) are undermined by the fact that single-node parallelism is not explored, benchmarked or optimised for.
  *As noted in the introduction paragraph, we have mixed up our computing terminology and have revised the paper to indicate that we are computing on "cores" rather than "processors." We have corrected all instances of this, as we compute on thousands of cores that are housed within various nodes on the computing architecture.*

- Section 2.3: The test hardware configuration is less than ideal for the given analysis, as the test nodes vary significantly ("8-24 cores with clock speeds ranging from 2.5GHz-3.06GHz"). For a rigorous performance analysis a fixed subset or partition of nodes with identical core types should be used, and the CPU model, as well as available memory needs be stated here.
  *We have included the available memory in the hardware section in lines 171-172. We also emphasize the ensemble process that we employ to combat the effects of the heterogeneous architecture in lines 173-175. These efforts include ensemble averaging for every core number—including serial runs.*

- While I appreciate that availability may be a limiting factor here, industrial compilers (Intel / Cray / Nvidia) are likely to achieve better single-node performance than gfortran -O3. If available on the test system, investigating reporting whether or not that changes the baseline (single core / single node) runs would add weight to the performance investigation. This can/should be considered optional.
  *We have considered comparing different compilers for these simulations, and it is likely that we will do so in the future. However, the availability of computational resources and the simulation data that we have already compiled make regenerating all simulations unlikely for this manuscript.*

- As clearly stated in section 5, accesses to particle of a neighbouring processors still go through a full MPI-driven halo exchange protocol, thus likely incurring significant memory movement that could further be optimised. Without appropriate treatment of shared memory parallelism, either via OpenMP or p-threads, or explicit MPI shared memory programming (one-sided) this is likely to incur significant

performance overheads that should be accounted for and evaluated in scalability studies such as this.

*I believe that this concern is still stemming from the processor/shared-memory mix up. Each core is responsible for the particles that reside within its assigned, local subdomain boundaries. Without implementing some OpenMP + MPI combination, the MPI-driven halo exchange must be conducted between cores to receive relevant nearby particles along neighboring subdomain boundaries. However, this manuscript provides the techniques that are necessary for a future version to implement inter-node communication in the same manner as we perform herein, laying the foundation for an OpenMP + MPI option. Once particles have been received, the K-D tree is constructed once on each core to be searched by each particle. In this way, we do not believe that the algorithm contains any redundant or unnecessary communication.*

- Section 6.1 Cost analysis: This seems like a rather lengthy way to explain and derive commonly known scaling behaviour (linear weak scaling with constant per-processor work and KD-tree lookup scaling as $nlog(n)$). I wonder if the manuscript can be made more concise by stating these facts and offering evidence that indeed the implementation behaves as expected?

  *This is a good point. However, it should be noted that we did not know that the MT/K-D tree routines, rather than the MPI/communication routine, would be the dominant cost prior to performing the scaling analysis. Rather, we discovered that these processes were dominant and then employed the established facts about the routines.*

- Figure 6: While Fig. 6 (a) does appear to match the predicted growth very well, the slope in Fig. 6 (b) is arguably steeper than the prediction. Could this be clarified or commented on?

  *This is an interesting observation. Although we haven't fully explored 3-d as much as we have 2-d, we believe that this is a cost of dimensionality, stemming from extra computations in the mass transfer and K-D tree routines. For example, the cost of searching a K-D tree after construction, $\mathcal{O}(N \log(N))$, does not depend on dimension, whereas the cost of building a tree structure is $\mathcal{O}(d * N)$ for each layer. We intended to provide and analyze a simple model that produces a basis for accurate speedup scaling. As seen in Figure 13, the model predicts 3-d speedup quite well, despite the difference in Figure 6.*

- Line 335-338; this concept is commonly referred to as "weak scaling" and should be named as such.

  *We believe that this concept has similarities to weak scaling, but what we are referring to here is not quite the same thing—at least not in the way we are describing the comparison. In this excerpt, we point to the fact that $N_S$ changes with the number of processors. So, in two totally different simulations with different **total** particle numbers, the value of $N_S$ can coincide for different numbers of cores, and we expect the same wall-clock time when that happens.*

- Figure 8 (omitted): The x-scale should be inverted. Furthermore, this figure does not actually add any value to the discussion, as the inverse relationship between total particles and local particles is trivial.

  *We agree that the figure creates confusion, and we have chosen to remove it and its discussion for clarity and fluidity.*

- Similarly, Fig 7(a). shows multiple instances of weak-scaling snapshots that are nowhere near the regime limits. Please consider replacing Fig 7(a). and Fig. 8(a) with an actual strong-scaling graph (wall time vs. number of cores) for total particle values chosen in Fig 7(a).

*We agree that this would be more familiar to computational readers, so we have added strong scaling graphs for each of the 2-d and 3-d runs on which we focus.*

- Figure 10: It is increasingly hard to think about scaling behaviour with an implicit number of processors. Please plot wall time against number of cores when assessing strong scaling!

  *As noted in previous comment, we have added the requested plots.*

- Fig 10: Are the drop-off points related to scaling beyond a single memory socket? The near-linear scaling beyond that point (counter-intuitively to the left!) suggests that scaling behaviour is memory-bandwidth limited and the addition of memory-sockets determines the real scaling factor. But again, this is hard to assess accurately without appropriate strong-scaling graphs.

  *This is a valid concern, and something that we are attributing to the condo-style, heterogeneous computing architecture available to us. In Figure 10, the fewest number of cores used is 100, which implies that we are past the point of inter-node communication playing a role in such a drop off. However, we do believe that it could be something related to communication between certain types of nodes on the cluster.*

- Units of time on y-axis missing for almost all wall time plots.

  *All relevant plots have been adjusted for this.*

- Section 7.2; line 470: This highlights one of the major shortcomings of the paper! While the single-core base case does not enter the MPI routine, it does represent an appropriate baseline for single-node/socket shared memory scaling. The authors not only neglect to account for this, but instead raise the baseline to an arbitrary 100-core baseline. To improve the quality of the paper, I recommend doing a specific shared-memory scaling analysis to determine if indeed scaling within a single memory socket follows the expected trends, and/or if the redundant use of ghost exchanges within a shared memory space affects performance. Then, once a single-socket/single-node baseline is established, scaling behaviour across multiple nodes/sockets can be analysed, thus giving a more grounded baseline for the speed-up plots in Fig. 16.

  *Figure 12 is a strong scaling figure with a baseline of a single core, and we have updated the figure to include the serial run for clarity. We agree with the reviewer that some single-node analysis is necessary to further validate our results, so we have included Figure 9 as an on-node communication example to show that cores within the same node participate in the exchange protocol as expected. Corresponding discussion has been added in lines 354-360. As for the 100-core baseline, we agree that it is a bit arbitrary. However, it is sometimes necessary if the computing architecture is memory-bound in such large simulations, as with some of the 3-d simulations in Figure 17. We have reworked the discussion and motivation in Section 7.2 to eliminate confusion for the inclusion of these figures. Also, the figures' axes have been updated to reflect that they are displaying a non-traditional speedup metric.*

- Line 482 and onward: The technical term for "above perfect efficiency" is "super-linear speed-ups". The explicit explanation of equation (30) might also be superfluous; instead a more detailed explanation of how super-linear speed-ups are achieved would certainly elevate the paper.

  *We tried to avoid using "super-linear speedup," because we understand that we are comparing to the non-typical 100-core baseline. We do not wish to claim that this is some grandiose accomplishment*

*for the algorithm since we simply thought that these figures were interesting in the context of large-scale simulations. Though, we agree that using the proper terminology is certainly better, so we have corrected it!*

- Section 8 outlines the lack of shared-memory parallelism; the question on line 522 already indicates that the authors are aware of this issue. Unfortunately, to my mind, this issue impacts the validity of the findings, since the demonstrated scaling behaviour (and the "hours to seconds" claim") can easily be put down to an insufficiently optimised baseline run. As no attempt is made to quantify the performance of the implementation with respect to single-node performance this casts doubt over the importance of the findings.

  *As noted in a previous comment, we implement an explicit, MPI-driven procedure that treats each core in the simulation as a non-shared memory resource. However, this manuscript provides the foundation for performing similar inter-node communication. If we intend for this to be a production code at some point, we agree that some combination of OpenMP and MPI processes will almost certainly be faster.*

- Future consideration: Latency hiding by overlapping ghost communication with compute work; possibly across time steps. Not for this work, but as a follow-on optimisation.

  *This is an interesting consideration! It's possible that some staggering of communication and calculations could slightly improve the overall wall-clock time if a more complex simulation is imperfectly load balanced, as we will inevitably encounter moving forward. We will certainly consider this proposition in our future work!*

**List of Changes**

- All instances of "processor(s)" have been corrected to say "core(s)."

- Memory capacity of computing architecture has been added to lines 171-172 in Section 2.3.

- Strong scaling plots, Figures 7, 8, and 10, have been added, and corresponding discussion regarding strong/weak scaling has been added to the section and the captions.

- Singe-node scaling discussion has been added to lines 354-360 in the beginning of Section 6.2 as well as the addition of the corresponding scaling plot Figure 9.

- (Old) Figure 8 has been omitted.

- Figure 12 was updated to include the serial run on the plot.

- All units of time have been added to relevant plot axes.

- Section 7.2 has been revised to emphasize the motivation for inclusion of non-serial reference speedup plots.

- We have corrected instances of "above perfect efficiency" to "super linear speedup."

- Various other minor adjustments (not pertaining to specific reviewer comments) to discussion, phrasing, and terminology have been made for clarity throughout the manuscript, as seen in the Diff pdf provided.

---

## Author Response (AR2)

**Response**

We have performed Roofline analysis on our algorithm's baseline. We provide citations and briefly mention of the Roofline model in the "Simulation Parameters" section. However, the figure and bulk of the discussion are in Appendix A, where we provide insight for the motivation of using Roofline analysis and give citations to the tools that we employed to generate our results.